# Anesthetic Management for Awake Craniotomy Applied to Neurosurgery

**DOI:** 10.3390/brainsci13071031

**Published:** 2023-07-05

**Authors:** Grazia D’Onofrio, Antonio Izzi, Aldo Manuali, Giuliano Bisceglia, Angelo Tancredi, Vincenzo Marchello, Andreaserena Recchia, Maria Pia Tonti, Nadia Icolaro, Elena Fazzari, Vincenzo Carotenuto, Costanzo De Bonis, Luciano Savarese, Leonardo Pio Gorgoglione, Alfredo Del Gaudio

**Affiliations:** 1Clinical Psychology Service, Health Department, Fondazione IRCCS Casa Sollievo della Sofferenza, San Giovanni Rotondo, 71013 Foggia, Italy; 2Complex Unit of Anaesthesia-2, Fondazione IRCCS Casa Sollievo della Sofferenza, San Giovanni Rotondo, 71013 Foggia, Italy; antonioizzi1201@gmail.com (A.I.); a.manuali@operapadrepio.it (A.M.); giulianobisceglia@live.it (G.B.); angelotancredi2@virgilio.it (A.T.); vincenzomarchello@libero.it (V.M.); a.recchia@operapadrepio.it (A.R.); mariapiatonti@alice.it (M.P.T.); freddydegaudio@libero.it (A.D.G.); 3Complex Unit of Neurosurgery, Fondazione IRCCS Casa Sollievo della Sofferenza, San Giovanni Rotondo, 71013 Foggia, Italy; nadia.icolaro@gmail.com (N.I.); elena.fazzari87@gmail.com (E.F.); kino.23@virgilio.it (V.C.); costanzo.debonis@operapadrepio.it (C.D.B.); l.savarese@operapadrepio.it (L.S.); l.gorgoglione@operapadrepio.it (L.P.G.)

**Keywords:** awake craniotomy, monitored anesthesia care, dexmedetomidine, remifentanil

## Abstract

Our anesthetic technique proposed for awake craniotomy is the monitored anesthesia care (MAC) technique, with the patient in sedation throughout the intervention. Our protocol involves analgo-sedation through the administration of dexmedetomidine and remifentanil in a continuous intravenous infusion, allowing the patient to be sedated and in comfort, but contactable and spontaneously breathing. Pre-surgery, the patient is pre-medicated with intramuscular clonidine (2 µg/kg); it acts both as an anxiolytic and as an adjuvant in pain management and improves hemodynamic stability. In the operating setting, dexmedetomidine in infusion and remifentanil in target controlled infusion (TCI) for effect are started. The purpose of the association is to exploit the pharmacodynamics of dexmedetomidine which guarantees the control of respiratory drive, and the pharmacokinetics of remifentanil characterized by insensitivity to the drug. Post-operative management: at the end of the surgical procedure, the infusion of drugs was suspended. Wake-up craniotomy is associated with reduced hospital costs compared to craniotomy performed in general anesthesia, mainly due to reduced costs in the operating room and shorter hospital stays. Greater patient satisfaction and the benefits of avoiding hospital stay have led to the evolution of outpatient intracranial neurosurgery.

## 1. Introduction

A neurosurgical approach practiced on an awake patient, or an awake craniotomy, was developed as early as the nineteenth century for the removal of epileptic foci under local anesthesia [1]. More recently, thanks largely to the improvements in monitoring methods and the availability of new anesthetic agents [2], the indications of this operative modality have been extended to the execution of stereotaxic biopsies, the treatment of vascular lesions [3], the resection of tumor lesions involving areas of language [4], and to the exeresis of supratentorial neoformations localized in different cortical areas [5]. The neurosurgical interventions carried out with this purpose include, in the intraoperative phase, the cognitive mapping [6] and administration of ad hoc neuropsychological tests while the patient is fully awake and cooperating [7] in order to preserve their cognitive capacity, and, at the same time, to obtain maximum neurosurgical excision [8]. The anesthetic techniques proposed for awake craniotomy are the monitored anesthesia care (MAC) technique, with the patient in sedation throughout the intervention [9]; the asleep-awake-asleep technique, which involves a phase of general anesthesia with orotracheal intubation; laryngeal mask (LMA), followed by an awakening phase, and, once the mapping is finished, a final stage of general anesthesia [10]; and the two-step anesthesia technique, asleep-awake (AA) anesthesia. The latter mode involves a phase of general anesthesia followed by awakening for mapping, after which, a subsequent sedation can be planned for the completion of the intervention [11].

Our protocol consists of the MAC technique, which involves analgo-sedation by administering dexmedetomidine and remifentanil in a continuous intravenous infusion, allowing the patient to be sedated and in comfort, but contactable and spontaneously breathing [12].

## 2. Materials and Methods

### 2.1. Ethical Consideration

The procedure was conducted in accordance with the Declaration of Helsinki, and approved by the Institutional Review Board of Fondazione Casa Sollievo della Sofferenza, San Giovanni Rotondo, Italy (Code: ConsInf An/01, approved 6 November 2013). The details can be obtained in the protocol which also includes a description of the protection of data.

### 2.2. Critical Issues

−Patient selection: Adult patients of both sexes were considered to undergo a resection of neoplasms near the Broca area, in particular low and high grade gliomas. The only absolute exclusion factor considered was the refusal of the subjects to undergo the awake procedure. Other relative exclusion criteria were American Society of Anesthesiologists Classification (ASA class) ≤ 3; difficult airway assessment; risk of intraoperative complications (e.g., risk of epilepsy); sedation assessment failure risk; neurological disorders and treatments; psychic and/or psychiatric disorders; abuse of alcohol, drugs and other psychotropic substances; chronic pain and opioid use for the treatment of this; low pain tolerance.−Preoperative: The management of the strategy which is to be adopted in a multidisciplinary manner is essential. In addition, the anesthetic visit and the interview with the patient guarantee an adequate preliminary assessment of the clinical and anesthetic conditions. Finally, correct anxiolysis allows for greater compliance with the procedures that are to be performed in the operating room.−Locoregional anesthesia: Regardless of the anesthetic and surgical techniques used in all the studies carried out on awake craniotomy, the use of local anesthetics at the scalp level is essential [13]. The reason why they are used is one of the important points of this discussion.−Management of analgo-sedation: The anesthetic technique used is a MAC (monitored anesthesia care) in which we tried to tailor the dosage and pharmacological infusion as much as possible by exploiting the synergy between the α2-agonists and remifentanil thanks to their pharmaceutical characteristics [14].

Management of any complications: Despite the low incidence in this procedure, acute conditions may arise which can lead to the failure of the anesthetic technique or which, in any case, involve the need to intervene to avoid serious neurological sequelae. In particular, we are talking about the need to intubate the patient, and therefore conditions of cognitive decline, the onset of worsening dyspnea, hemodynamic alterations, the appearance of epileptic seizures, etc.

### 2.3. Procedure

#### 2.3.1. Pre-Operative Phase

The discussion of the case takes place in a multidisciplinary context with the involvement of a neurosurgeon, an anesthesiologist, and a psychologist; the anesthesiological visit and the psychological interview with the patient are then carried out (Table 1).

As part of the preoperative assessment, the neuropsychological framework is essential for the following:Assessing the degree of collaboration;Evaluating cognitive impairments;Presenting the tests to be submitted to the patient in the operating room.

From an anesthetic point of view, the patient undergoes an accurate clinical evaluation about a week before surgery. A thorough medical history and physical examination allow us to obtain useful information to better frame clinical cases. The data are confirmed by blood chemistry and instrumental tests based on the need for further study according to the current guidelines, and by any specialist consultations that best outline the pathophysiological contexts to be highlighted. Any potential risky situations to be faced in the operating room are prevented through the necessary precautions (presence of the cart for difficult intubations in cases where such an eventuality can be envisaged, high risk of onset of epileptic seizures in subjects suffering from this condition, or in which the area to be operated on is particularly at risk, etc.) [15]. The evening before the surgery, and half an hour before arriving in the operating block, the patient is pre-medicated with intramuscular clonidine (2 µg/kg); it acts both as an anxiolytic and an adjuvant in pain management and improves hemodynamic stability [16].

#### 2.3.2. Management in the Operating Room

Preparatory phase: The patient is re-evaluated upon arrival in the operating room, in the event that new morbid conditions may have arisen in the meantime (fever, hemodynamic changes, etc.). The last doubts or concepts that could transmit anxiety or fear to the patient are clarified, and then the patient is placed on the operating table. The monitoring of vital parameters is carried out by means of a standard monitor that allows the display of the values of SpO2, T, non-invasive blood pressure (NiBp), heart rate (HR), respiratory rate (RR), and state of consciousness. Arterial lines or urinary catheters were not routinely inserted. On the other hand, at least two large-caliber peripheral vascular accesses (16–18 G) were found. The continuous manual infusion of dexmedetomidine at low doses (0.7–1 µg/kg/h) and of remifentanil via target controlled infusion (TCI), in accordance with the Minto model (0.5–1 ng/mL), are started by means of an infusion pump. All patients breathed spontaneously and received supplemental oxygen at 4 L/min (inducing a FiO2 of about 36%) through a nasal cannula, increasing the inspiratory reserve, improving tissue oxygenation, and reducing the risk of accidental desaturations. Thereafter, analgesic blocks of the nerves of the scalp were performed using levobupivacaine or ropivacaine without exceeding the maximum dosage allowed for the patient’s weight and in relation to age. The blocks, in particular, were performed at the level of the six cranial nerves that innervate the scalp, hence the name scalp block, bilaterally distributing the local anesthetic [17]. The overall management of the anesthetic with respect to drug infusion adjustments was left to the anesthetist present. Afterwards, the most appropriate decubitus for the procedure was taken, taking into account both the surgical needs, and, above all, the comfort of the patient who due to their long stay in the same position could complain of pain, fidget, and move during the surgery, causing problems for the surgeon and even leading to lesions of brain structures, invalidating the procedure. Furthermore, anti-decubitus devices were placed in the areas most at risk of injuries of this type. Approximately 10 min later, with local anesthetic agent (2% lidocaine with 1:200,000 epinephrine), the pin placement sites of the Manfield fixator were infiltrated and then placed by the neurosurgeon to keep the head immobile during the procedure. It should be noted that this would not be possible after the locoregional analgesia of the affected area, as the bone tension caused would be too painful [18]. At any time during the procedure, when excessive pain was expected or if the patient complained of pain or discomfort, the infusion rates of dexmedetomidine and/or remifentanil were increased.

Awake craniotomy phase: Once the craniotomy is performed, the dura mater is opened and the neurosurgeon begins with the exposed tumor excision. As previously mentioned, during skin incision and dura mater manipulation, an increase in drug dosages may be necessary, especially for remifentanil. Afterwards, it may be necessary to reduce the doses of drugs in the continuous infusion as the brain parenchyma has a poor innervation of pain sensitivity and the patient must remain awake and cooperating in order not to invalidate the tests that allow for the cognitive-motor evaluation. The infusion rates are adjusted not only according to the needs of the procedure but also according to the patient’s requests, and vital parameters constantly monitored. Pain assessment is performed through the numerical rating scale (NRS). During the surgical excision, the patient is awake, spontaneously breathing and perfectly cooperating. The mapping of the affected brain areas takes place through the placement of stimulating electrodes on the cortical surface by the neurosurgeon. Thanks to this, it is possible to minimize lesions in areas not affected by the neoplasm, allowing the assessment of cognitive abilities and in particular the integrity of the areas of language and association [19]. Intraoperative neuropsychological testing is conducted by administering to the patient the cognitive tests already performed previously in the pre-operative phase. In addition to the administration of the tests, an overall neuropsychological evaluation is carried out by the psychologist, and the neurosurgeon determines the positive areas that will allow for establishing the limits of the lesion resection. During the intervention, the neuropsychologist and the anesthetist remain next to the patient until the neurosurgical intervention is completed.

Post excision phase: After removing the tumor mass and checking for any bleeding, the reconstruction and suturing phase begins. In this phase, the algogenic stimuli of the skin and brain membranes are considerable. Anesthetic management requires the dosages of the infusion drugs to be increased again, according to the clinical and respiratory hemodynamic parameters, to prevent the pain.

#### 2.3.3. Post-Operative Phase

Postoperative management: At the end of the surgical procedure, the infusion of drugs was suspended and the patient was placed back in the supine position and transported to the recovery room. The vital parameters were constantly monitored and the GCS (Glasgow Coma Scale) and the Aldrete scale (Figure 1) were evaluated. If there was an onset of side effects (vomiting, nausea, shivering, pain, anxiety, epileptic seizures, breathing difficulties, hemodynamic changes, etc.), they were resolved immediately. Patients stayed for about an hour in this room, and, after a final evaluation of vital and cognitive parameters, they were discharged and transported to the departments of origin. Postoperative pain was treated with paracetamol 1 g every 8 h, max 4 g/day. Ondansetron 4 mg or metoclopramide 20 mg and/or dexamethasone 4 mg i.v. were administered for postoperative nausea and vomiting when needed. In the ward, the patient was monitored by the staff and reassessed at least once every hour. Discharge, unless complications occurred, was within 1–2 days of surgery.

## 3. Outcome Measures

The outcome measures were the following:−Width of the tumor area removed;−Reduced intra- and post-procedural pain;−Patient gradation;−Grade of the surgeon;−Reduced costs;−Fewer hospital stays;−Reduced complications compared to general anesthesia;−Reduced side effects.

## 4. Discussion

Neurosurgical interventions conducted with the awake craniotomy approach, applied to the treatment of tumor lesions, allow the safeguarding of the patient’s cognitive abilities.

However, for the success of this technique, close collaboration is required between the various professionals involved (neurosurgeon, anesthetist, neuropsychologist, and nurse). With reference to anesthetic management, the anesthesiologist’s challenge is to develop a technique that allows for safely obtaining an awake state to ensure that the patient is able to cooperate and perform neuropsychological tests.

Consequently, it is mandatory to refer to a detailed anesthetic protocol, making the most of the pharmacokinetic/dynamic characteristics of the drugs used, and the operating methods of the drug administration and anesthesia monitoring systems [20].

We can consider a clear advantage in using anesthetic management of this type, as the possibility of communicating with the patient and keeping in contact with them during all phases of the procedure is an invaluable factor compared to different types of management in which the patient is not contactable in some moments.

Surely, all of this is guaranteed by the synergism of dexmedetomidine in infusion and remifentanil in TCI for effect. The purpose of the association is to exploit the pharmacodynamics of dexmedetomidine, which guarantees the control of respiratory drive and the pharmacokinetics of remifentanil, characterized by insensitivity to the drug [21].

Conscious sedation with dexmedetomidine has several advantages over the use of propofol in continuous infusion, including less demand for vasoactive agents, reduced opioid consumption, better compliance with intraoperative tests, and a shorter duration of surgery and hospital stay [22].

Sedation with dexmedetomidine is particularly useful in high-risk patients or when surgery is likely to be prolonged due to its ability to reduce postoperative delirium and cognitive impairment, and postoperative cognitive dysfunction (POCD) [23]. Fundamental research suggests that dexmedetomidine converges with a natural sleep pathway to exert its sedative effect. This unique state of sedation, also called “collaborative” sedation, can be useful for AC, which requires a deep level of sedation during painful and stimulating operating procedures on the one hand, and sufficient patient cooperation during eloquent function mapping on the other [24].

In this scheme, the use of clonidine in premedication contributes not only to the preoperative control of surgical stress through anxiolysis, in perfect compliance with the Enhanced Recovery After Surgery (ERAS) protocol [25], but also to a condition of preemptive analgesia, favoring opioid sparing and the action of local anesthetics during the execution of nerve blocks of the scalp [26].

Precisely regarding this point there is an even greater advantage, namely that local anesthesia avoids not only pain during the surgical cut, but above all the distress during the positioning of the pins of the Mayfield clamp, which certainly involves strong bone tension. There is a large variation in the occurrence of postoperative pain and the need for analgesic drugs in neurosurgical patients [13].

The experience of the surgeons naturally reduces the risks associated with excessive bleeding, and therefore significant cardiovascular alterations and conditions related to the triggering of epileptic seizures and altered epileptic foci, which inevitably involve the use of drugs that can modify the anesthetic’s plan.

A psychologist’s support for this type of patient, who already has a dramatic family and social condition and experience, is very important. Psychologically preparing a patient for all phases of such an invasive surgical procedure, or simply for anything that involves being in an operating room and experiencing all of the lights, sounds, noises, and smells in the midst of surgery can be a nightmare. This is why the presence of a professional figure that can minimize stress and discomfort both pre- and intra-procedurally is fundamental, and impossible to eliminate.

In addition, it is necessary to consider the patient tests performed during the resection phase of the area adjacent to Broca’s, in which an evaluation of the results is required to consider the integrity of language, visual association, and data processing skills. Intraoperative monitoring is essential, as alterations in vital parameters can help to prevent any ongoing decompensation before clinical events of instability occur.

The efficacy of intraoperative seizure prophylaxis with anticonvulsants remains questionable; levetiracetam may be superior to other drugs for this purpose [27,28]. Awake craniotomy is associated with a shorter length of hospital stay compared to craniotomy under general anesthesia, potentially reducing the risks of nosocomial infection and thromboembolic complications, and shortening the intervals between surgery and the initiation of chemotherapy and/or radiotherapy if required.

Wake-up craniotomy is associated with reduced hospital costs compared to craniotomy performed with general anesthesia, mainly due to reduced costs in the operating room and shorter hospital stays.

Greater patient satisfaction and the benefits of avoiding hospital stay have led to the evolution of outpatient intracranial neurosurgery [29,30].

## 5. Conclusions

To conclude, we would first like to underline how much work it takes to be able to formulate a valid and complete protocol in the peri-operative management of a patient who is preparing to undergo awake craniotomy. The coordination of the team and the preoperative preparation of the management of each patient are the keys to obtaining the best result, both from an anesthetic and a surgical point of view. Furthermore, it must be said that it is not easy to recruit a patient to undergo this type of procedure, and who will agree to stay awake; it is important to remember that the first request that any type of patient normally makes is that they want to sleep and not see or feel anything until they wake up. We can affirm that the synergy between opioid drugs and alpha-2 agonists is efficacious in allowing the necessary comfort of the patient and the correct mapping of the area to be treated. The problems with the optimal anesthetic support of awake craniotomy for high-risk patients with somatic or neurological disorders are central to future research. Although brain tumor surgery remains the most common indication for awake craniotomy, the technique is finding use in other neurosurgical procedures.

## Figures and Tables

**Figure 1 brainsci-13-01031-f001:**
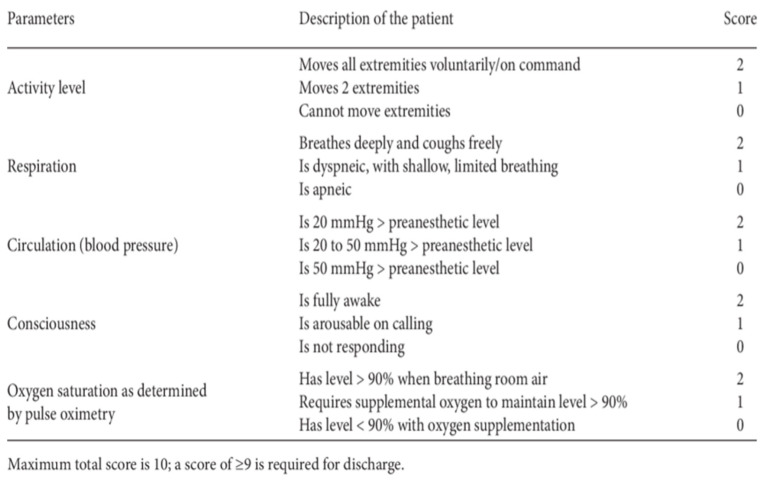
Aldrete score.

**Table 1 brainsci-13-01031-t001:** Patient selection criteria.

Age > 18
ASA class (≤3)
Airway assessment
Risk of intraoperative complications (e.g., risk of epilepsy)
Failure risk of sedation assessmentNeurological disorders and treatment
Psychic and/or psychiatric disorders
Abuse of alcohol, drugs, or other psychotropic substances
Chronic pain and opioid use for treatment
Low pain tolerance

## Data Availability

Not applicable.

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
