# Peer review of "Anesthetic Management for Awake Craniotomy Applied to Neurosurgery"

_brainsci, 2023, doi:10.3390/brainsci13071031_

Round 1
Reviewer 1 Report
Reporting of protocols should be as per SPIRIT guidelines. the manuscript needs extensive correction based on above guidelines. many importnatmparts are missing : aims and objectives, trial design, randomization, blinding, allocation concealment, sample size calculation, intervention groups.

Author Response
Thanks to the Reviewer for his/her observation. We clarify that the manuscript is a procedure already routinely used in our hospital for patients who undergo awake surgery craniotomy. Our manuscript is not a clinical trial. We want to tell and share our experience about the anesthetic procedure during a particular operative circumstance such as awake surgery.
Reviewer 2 Report
The authors describe protocol for awake craniotomy, but not a clinical trial. The authors submitted the manuscript as an article. I do not think the manuscript is a clinical trial. The authors do not show any hypothesis, patients, or data.
None
Author Response

(The authors gave the same response as above.)

Round 2
Reviewer 2 Report
Honestly, I do not understand why the authors submitted the manuscript. This is not a research article. Are the authors trying to submit review or protocol? Please clarify the article type.
None.
Author Response
Honestly, I do not understand why the authors submitted the manuscript. This is not a research article. Are the authors trying to submit review or protocol? Please clarify the article type.
According to the reviewer observation, we clarified that the manuscript is a protocol on the header of the sheet.
Round 3
Reviewer 2 Report
I reviewed the manuscript. My review comments are below.
Comments and Suggestions for Authors
I believe that the manuscript is acceptable if extensive English editing is done. Also, I have several minor comments.
Minor comments
#1. It appears that the English check of the present version has been done by a person not familiar with the contents, and there are numerous sentences that are grammatically correct, but have awkward meaning.
#2. I do not think informed consent is necessary in protocol (Page 2, Line 64).
#3. I do not think Figures 1 and 2 are necessary.
#4. Please discuss one theme in one paragraph in the discussion. First three paragraphs are too short and fourth is too long.
#5. Conclusion is redundant. Please shorten.
It appears that the English check of the present version has been done by a person not familiar with the contents, and there are numerous sentences that are grammatically correct, but have awkward meaning.
Author Response
I reviewed the manuscript. My review comments are below.
Comments and Suggestions for Authors
We thank the Reviewer for his/her thoughtful and constructive comments. We have addressed each of the issues raised and have highlighted the relevant revisions in the manuscript itself. Below, please find item-by-item responses to the Reviewer’s comments, which are included verbatim.
I believe that the manuscript is acceptable if extensive English editing is done. Also, I have several minor comments.
According to reviewer suggestion, we re-obtained a review of the manuscript from a native English speaker.
Minor comments
#1. It appears that the English check of the present version has been done by a person not familiar with the contents, and there are numerous sentences that are grammatically correct, but have awkward meaning.
- As suggested by reviewer, an anesthesiologist and a neurosurgeon reviewed the text.
#2. I do not think informed consent is necessary in protocol (Page 2, Line 64).
- We deleted the informed consent citation at Page 2, Line 64, as recommended.
#3. I do not think Figures 1 and 2 are necessary.
- We deleted the aforesaid figures.
#4. Please discuss one theme in one paragraph in the discussion. First three paragraphs are too short and fourth is too long.
- In the Discussion section, we distributed the paragraphs, as recommended.
#5. Conclusion is redundant. Please shorten.
- According to reviewer, we have summarized the Conclusion section, deleting redundant sentences.